# Evidence of Community-Wide Spread of Multi-Drug Resistant *Escherichia coli* in Young Children in Lusaka and Ndola Districts, Zambia

**DOI:** 10.3390/microorganisms10081684

**Published:** 2022-08-21

**Authors:** Flavien Nsoni Bumbangi, Ann-Katrin Llarena, Eystein Skjerve, Bernard Mudenda Hang’ombe, Prudence Mpundu, Steward Mudenda, Paulin Beya Mutombo, John Bwalya Muma

**Affiliations:** 1School of Medicine, Eden University, Lusaka P.O. Box 37727, Zambia; 2Department of Disease Control, School of Veterinary Medicine, University of Zambia, Lusaka P.O. Box 32379, Zambia; 3Faculty of Veterinary Medicine, Norwegian University of Life Sciences, 1432 Ås, Norway; 4Department of Paraclinical Studies, School of Veterinary Medicine, University of Zambia, Lusaka P.O. Box 32379, Zambia; 5Department of Environmental and Occupational Health, Levy Mwanawasa Medical University, Lusaka P.O. Box 33991, Zambia; 6Department of Pharmacy, School of Health Sciences, University of Zambia, Lusaka P.O. Box 50110, Zambia; 7Kinshasa School of Public Health, Faculty of Medicine, University of Kinshasa, Kinshasa 834, Congo

**Keywords:** antimicrobial resistance, *Escherichia coli*, children, risk factors, Zambia

## Abstract

Increased antimicrobial resistance (AMR) has been reported for pathogenic and commensal *Escherichia coli (E. coli)*, hampering the treatment, and increasing the burden of infectious diarrhoeal diseases in children in developing countries. This study focused on exploring the occurrence, patterns, and possible drivers of AMR *E. coli* isolated from children under-five years in Zambia. A hospital-based cross-sectional study was conducted in the Lusaka and Ndola districts. Rectal swabs were collected from 565 and 455 diarrhoeic and healthy children, respectively, from which 1020 *E. coli* were cultured and subjected to antibiotic susceptibility testing. Nearly all *E. coli* (96.9%) were resistant to at least one antimicrobial agent tested. Further, 700 isolates were Multi-Drug Resistant, 136 were possibly Extensively-Drug Resistant and nine were Pan-Drug-Resistant. Forty percent of the isolates were imipenem-resistant, mostly from healthy children. A questionnaire survey documented a complex pattern of associations between and within the subgroups of the levels of MDR and socio-demographic characteristics, antibiotic stewardship, and guardians’ knowledge of AMR. This study has revealed the severity of AMR in children and the need for a community-specific-risk-based approach to implementing measures to curb the problem.

## 1. Introduction

The emergence of antimicrobial resistance (AMR) is a global public health threat [1]. Increased resistance to commonly used antibiotics has been reported for various pathogenic and commensal bacteria, hampering the treatment, and increasing the burden of infectious diarrhoeal disease in general, especially in children under five years [2,3]. Resistance against all available antimicrobial agents has been reported [4], including the last line of antibiotics reserved for treating infectious diarrhoeal diseases, which is medically alarming [5].

In developing countries, the burden of childhood infectious diseases remains high [6], partly due to inadequate health care systems [7]. Infectious diarrhoeal disease is one of the significant causes of mortality and morbidity in children under-five years in developing countries, including Zambia [8,9]. Pathogenic *Escherichia coli* is a major cause of bacterial diarrhoeal disease in this age group [10] and AMR increases the burden of such diarrhoeal diseases even further [3].

The rapidly growing trend of AMR is a multifaceted problem driven by several interlinked factors, including inherent microbial characteristics, selective pressures of antimicrobial use, and changes in society and technology that enhance the transmission of drug-resistant organisms [11,12]. A significant driver of AMR development is the over- and misuse of antimicrobials as therapeutics in human and veterinary medicine, agriculture growth promotors, and disinfectants in households [13,14,15,16]. Many of these compounds end up in the environment [17,18,19], hence contributing to the spread of AMR in animals, the environment, and directly or indirectly to humans [20,21,22,23]. Further, other factors such as behavioural, e.g., self-prescription of antibiotics [24]; sanitation and demographic e.g., crowded settings, poor cleanliness [25,26]; socio-economic e.g., poverty [27,28], have been implicated in the spread of AMR in communities. 

Increased consumption of antibiotics during the last decades, especially in developing countries [29] has led to an increase in the occurrence of AMR [30,31]. Due to the lack of strict regulations on the use of antibiotics in many developing countries, the population has easy access to these compounds, even without prescription [32,33,34]. In addition, despite the World Health Organization (WHO) recommendation to reserve antibiotics only in cases of bloody diarrhoea [35], antibiotics are readily used to treat any form of diarrhoea in children [36].

Most studies have focused on AMR in pathogenic *E. coli* collected from diarrhoeic children, as antimicrobial susceptibility patterns govern the treatment and AMR pathogenic bacteria is a direct public health threat. However, samples from healthy children may reveal resistant commensal *E. coli* that might act as significant reservoirs for resistance genes [37] hence, playing a considerable role in spreading the resistance within a community [38]. Tenover and McGowan (1996) suggested that exposure of commensal bacteria like *E. coli* to antibiotics increases the carriage levels of resistant organisms and, if plasmid-mediated, resistance might be transmitted to a more virulent acquired organism [39]. 

Therefore, this study aimed to explore the occurrence and patterns of antimicrobial-resistant *E. coli* isolated from children under five years old in Zambia and identify possible drivers for AMR in the study population. We used *E. coli* since it is commonly found in humans and animals, can cause diseases in both host categories, and might serve as markers of antibiotic resistance spread to pathogens and the remaining gut microbiota [40,41].

## 2. Materials and Methods

### 2.1. Study Design, Sites, and Population

A hospital-based cross-sectional study was conducted in 12 purposively selected health centres (hospitals) and the children’s hospital in Lusaka and Ndola districts, respectively. The study sites are the provincial headquarters of the most populated provinces of Zambia and host heterogeneous populations from different cultures and social backgrounds [42] (Figure 1).

### 2.2. Sample Size and Sampling Strategy

The population of children under five years of age in the Lusaka and Ndola districts in 2019 were projected at 425,000 and 89,000, respectively [43]. If 50% of these children sought health services [44], a fraction of 0.5% would be considered representative, and the chosen sample size would be large enough to detect rare varieties of AMR. The targeted sample size was, therefore, estimated to be 1287 children (diarrhoea and non-diarrhoeic). Sampling was proportionally distributed in both districts using the under-five population as a weighing proxy factor. The standard World Health Organization (WHO) definition of diarrhoea was used [45], while healthy children were those without any symptomatic disease at the time of visiting the clinic. Children undergoing antibiotic treatment at the time of sampling and non-concerting parents were excluded.

### 2.3. Sample Collection and Epidemiological Survey

A rectal swab specimen was aseptically collected from each study participant and transported chilled (<4 °C) in Cary-Blair enteric transport media (Oxoid, Basingstoke, UK) to the Bacteriology Laboratory at the University Teaching Hospital in Lusaka for analysis. Health status, food habits, socio-demographic characteristics, antibiotic stewardship, and awareness of AMR of guardians were investigated through a pre-tested structured questionnaire administered to the child’s guardian. The final questionnaire used is provided as a Appendix A.

### 2.4. Laboratory Analysis

#### 2.4.1. Isolation and Identification of *E. coli*

Rectal swabs were pre-enriched in buffered peptone water (Oxoid, Basingstoke, UK) and aerobically incubated at 37 °C for 24 h. The enriched broth was plated onto MacConkey agar plates (Oxoid, Basingstoke, UK) and incubated aerobically for an additional 24 h at 37 °C. Lactose fermenting colonies were then sub-cultured onto Eosin Methylene Blue (EMB) agar plates (Oxoid, Basingstoke, UK) and incubated aerobically at 37 °C for 24 h. Presumptive *E. coli* colonies displaying a green metallic sheen were then purified on nutrient agar (Oxoid, Basingstoke, UK). One colony from each plate was further confirmed by phenotypic characterization and standard biochemical tests using triple sugar iron (Oxoid, Basingstoke, UK), Sulphur Indole Motility (Oxoid, Basingstoke, UK) and citrate agar (Oxoid, Basingstoke, UK). For additional taxonomic confirmation, 323 isolates were randomly selected for further identification by matrix-assisted laser desorption ionization time-of-flight mass spectrometry (MALDI-TOF MS) using the VITEK^®^ MS—SARAMIS^®^ KB V4.16 (bioMérieux, Lyon, France).

#### 2.4.2. Antimicrobial Susceptibility Testing (AST)

The AST was performed by the Kirby-Bauer disc diffusion method using the Clinical Laboratory Standards Institute (CLSI) guidelines on Müeller-Hinton agar plates (Oxoid, Basingstoke, UK) [46]. Suspensions of 0.5 McFarland were prepared from pure colonies of isolated *E. coli* and inoculated onto Müller-Hinton agar plates (Oxoid, Basingstoke, UK). The susceptibility pattern of the isolates was determined for a panel of ten (10) antibiotics (Table 1). A standard culture of *E. coli* (ATCC 25922) was used as positive control culture with each batch of antimicrobial susceptibility testing. 

The plates were incubated for 16–18 h at 37 °C. The zones of inhibition were read using a digital Vernier Calliper and interpreted as Susceptible (S), Intermediate (I), and Resistant (R) based on the CLSI guidelines [46]. Multi-Drug Resistant (MDR), Extensively-Drug Resistant (XDR), and Pan-Drug-Resistant (PDR) isolates were identified, with MDR defined as non-susceptibility to at least one antibiotic in three antimicrobial classes tested; XDR as non-susceptibility to at least one antibiotic in all but two or fewer antimicrobial classes (i.e., *E. coli* isolates remain susceptible to only one or two classes); PDR as non-susceptibility to all antibiotics in all antimicrobial classes tested [47]. Since only one antimicrobial agent was tested for each antimicrobial class, the concepts possible XDR and possible PDR were used as per the international expert proposal for interim standard definitions for resistance recommendations [47]. 

### 2.5. Data Analysis

The measured diameters of the zones of inhibition for AST were analysed using the WHONET 2021^®^ software. The resistance profile for all antibiotics was reported, and tables and graphs were produced in WHONET. Epidemiological data and certain outputs from WHONET^®^ 2021 software were summarized and then entered into a database using Excel 2016^®^. Further statistical analyses were completed using Stata (StataCorp, College Station, TX) version 16.0 for Windows. 

Initially, descriptive statistics focused on describing the categorical variables from the questionnaire focusing on demographic and hygienic factors, as shown in Table 2 and Table 3. The potential associations between the hypothesized categorical risk factors and the dichotomous outcomes (MDR, XDR, and PDR patterns displayed by *E. coli*) were assessed using Chi-square analyses. A new variable was created as an ordinal outcome with 0 = AMR, 1 = ANY AMR, 2 = MDR, 3 = XDR, and 4 = PDR. Explanatory variables showing a *p*-value < 0.20 from one of the outcome models were selected as candidate variables and taken into the multivariable logistic and ordinal regression models. The multivariable models were built using a backward selection strategy, using a *p*-value of <0.05 of the likelihood ratio test as inclusion criteria. The model fit was assessed using the Hosmer Lemeshow test, *lroc* and *lsens* procedures in Stata for logistic models, and graphical methods for the ordinal model. Finally, the potential effects of the random effect of Health Centres were assessed for all models using the *melogit* procedure in Stata.

### 2.6. Ethical Consideration

The study was approved by the Zambian ERES Converge Institutional Review Board (Ref. No. 2020-Aug-006) and the National Health Research Authority (NHRA00010/3/09/2020). Further, the Provincial and District Health Offices were informed about the study. Informed consent was obtained before the study, and only the under-five-year-old children, as defined above, whose guardians consented to participate in the study were included.

## 3. Results

### 3.1. Characteristics of Study Participants

In total, 1020 children were included in this study, of which 565 (55.39%) were diarrhoeic and 455 (44.61%) healthy. Boys were slightly more represented than girls (*n* = 521, 51.08%). The median age (IQR) of study participants was 10 (4–21) months, with the minimum and maximum age range between one and 59 months, respectively. Most of the study participants lived in high-density population areas (77.84%), and 58.92% of their guardians had attained up to a secondary level of education. Further, more than half, 598 (58.62%), of the households had five or more members (Table 2).

With regards to the variables connected to hygiene and behavioural characteristics, 859 (84.22%) households used council water (pipe-borne), and 439 (43.04%) further treated water intended for drinking. Most guardians, 728 (71.37%) and 783 (76.76%) reported washing their hands before cooking or feeding the child and after disposing of the child’s faeces, respectively. The pit latrine was used slightly more (52.55%) than the flush toilet, while the bin (73.43%) was the commonly used disposal method of solid waste. Only 191 (18.73%) of the participant’s guardians complied with the exclusive breastfeeding of children below six months. Most guardians, 736 (72.16%), stored prepared food for children for further use, mostly at room temperature (44.97%) or in a warmer (43.75%), and the spoon (72.16%) was commonly used to feed children (Table 3).

The knowledge and characteristics of antibiotics, AMR, and diarrhoea are summarized in Table 4. Most guardians, 531 (77.75%), perceived the consumption of contaminated food as the cause of their children’s diarrhoea. Diarrhoea with mucus 301/565 (53.46%) and fever 347/565 (61.63%) were the most frequently reported symptoms in children with diarrhoea. Further, most guardians reported that they lacked knowledge of antibiotics and AMR. Of these, 110/175 and 26/37 correctly understood antibiotics and AMR, respectively. 

### 3.2. Antimicrobial Susceptibility Patterns

The identification of isolates using standard biochemical tests was reliable since their confirmation using MALDI-TOF MS showed 98.8% accuracy. All *E. coli* isolates were subjected to AST using the above-mentioned panel of antibiotics (Table 1). Most *E. coli* were resistant to at least one antibiotic (988/1020), nearly equally distributed between healthy and diarrhoeic children with 95.4% (434/455) and 98.1% (554/565), respectively (S1). The *E. coli* isolates displayed the highest resistance against ampicillin (78.0%), trimethoprim-sulfamethoxazole (70.4%), and tetracycline (62.8%), while they were more susceptible to chloramphenicol (83.8%) and gentamicin (80.1%) (Figure 2a). This trend was the same in both healthy and diarrhoeic children, although at different percentage levels. However, a significant difference in the susceptibility and resistance profiles of the isolates to imipenem was observed between the healthy and diarrhoeic children. Nearly 62% and 24% of isolates from healthy and diarrhoeic children were resistant to imipenem, respectively (Figure 2b).

Of the 1020 *E. coli* isolates, 82.8% (845) were MDR, with 136 and 9 isolates being possible XDR and possible PDR, respectively. The MDR profiles were grouped in 251 different patterns consisting of 208 MDR patterns (resistance to at least one antibiotic in three antimicrobial classes tested), 41 possible XDR patterns (resistance to at least one antibiotic in all, except in one or two, antimicrobial classes tested) and two possible PDR patterns (resistance to all antibiotics in all antimicrobial classes tested) (Appendix A). The five most frequent MDR and possible XDR patterns occurred differently between both groups, with patterns A and D being significant for MDR (Figure 3a). The frequency of the five possible XDR patterns was significantly different between the groups (Figure 3b), while possible PDR *E. coli* were isolated from healthy children only.

### 3.3. Potential Risk Factors Associated with AMR

The dichotomous outcome variables MDR, possible XDR, and possible PDR were first analysed individually and merged into a single ordinal outcome variable called Levels of AMR (LAMR). Each outcome variable was analysed into three categories, diarrhoeic, healthy, and all children. In the univariable analysis, all variables with a *p* < 0.20 (Table 5) were selected to build the multivariable logistic regression and ordered logistic regression models.

Results from the standard multivariable logistic regression analysis are shown in Table 5. Different sets of variables emerged in the different models, and many variables were removed when adjusting for the random effect of health centres. All the logistic regression models adequately fit the data (Hosmer and Lemeshow test), but with limited explanatory power, with ROC areas around 0.60 (a ROC area of 0.50 indicated no explanatory power). Substantial model improvements were observed for the random effect (LR test, *p* < 0.001).

In the MDR-adjusted models, only storing prepared food for the child was significantly linked to MDR in all children, while gender and guardians’ awareness of antibiotics remained in the healthy children model. For XDR, after adjusting for the effect of health centres, five variables were removed, while age groups 6 to 11 and 12 to 35 months became significantly associated with possible XDR isolates (Table 5). For PDR, the only variable remaining for healthy children was awareness of AMR.

Table 6 shows results for LAMR, the combined ordinal variable. While several factors were identified in the standard multivariable model, adjusting for clustering removed most variables from the model. Only feeding a child with stored prepared food remained.

## 4. Discussion

This study investigated the occurrence and patterns of AMR—*E. coli* in healthy and diarrhoeic children and established risk factors for AMR in children below five years. Nearly all *E. coli* (96.9%) were resistant to at least one antimicrobial tested, slightly above the 83% reported in Nigeria [48], and resistant *E. coli* were equally common among healthy and diarrhoeic children. This near absence of susceptible *E. coli* is an alarming sign, foreshadowing a future public health crisis. Indeed, AMR is frequently observed in many low- and middle-income countries, as reported in Ethiopia [49], Bolivia, Peru [50], and Vietnam [51], including in children’s commensal *E. coli* in Kenya [52]. Some studies on commensal *E. coli* isolated from children from Asia found a lower occurrence of resistance [51,53]. However, these studies were conducted in rural areas, which could explain their lower reporting rates of AMR as access to and therefore misuse of antibiotics are more prominent in urban areas [54,55]. Equally, resistant *E. coli* isolated from diarrhoeic children were common in Burkina Faso [56], South Africa [5], and Ethiopia [57] corroborating our findings, while relatively different trends were observed in Taiwan [58] and Pakistan [59]. The difference in the setting of these two studies could justify this disparity. 

Several studies have reported high resistance to commonly used antibiotics, e.g., ampicillin, trimethoprim-sulfamethoxazole, and tetracycline, against enteric bacterial infections [49,58,59,60]. The easy accessibility and affordability of these drugs have led to their overuse by the population [29,33] but also in the animal production chain, as evidenced by recent studies completed in Zambia [22,23,61]. Further, in countries with a high prevalence of HIV infection, like Zambia [62], trimethoprim-sulfamethoxazole has been heavily used for the past decades as prophylaxis against opportunistic infections in HIV-infected and/or exposed individuals [63,64]. The above factors could explain the maintenance of this resistance through selection pressure. 

We found that most *E. coli* were susceptible to chloramphenicol and gentamicin. An earlier study from Zambia [65] also found a low prevalence of chloramphenicol and gentamicin-resistant *E. coli* isolates. This observation implies that the strains have not developed more resistance against these two drugs since the last study in Zambia by Chiyangi [65]. However, increased resistance to antibiotics recommended in the Zambia standard treatment guidelines for infectious diseases in children, including diarrhoeal diseases [66], limits treatment options. 

There were few observable differences in resistance patterns between *E. coli* in healthy and diarrhoeic children. Surprisingly, *E. coli* from healthy children were more frequently resistant to commonly used antimicrobial like amoxicillin-clavulanic acid, ciprofloxacin, and gentamicin compared to diarrhoeic children. This points to the community-level circulation of resistance as opposed to hospital-acquired resistance or resistance associated with disease management. Of special worry was the high percentage of imipenem-resistant isolates observed in healthy children; imipenem-resistant *E. coli* were almost three times more commonly isolated from healthy compared to diarrhoeic children. The high prevalence of resistant strains against imipenem is not unique to this study [5]. It is especially worrisome as the drug is a last resort classified as critically important and a high-priority antimicrobial agent to treat severe bacterial infections in humans [67]. This scenario supports previous postulates that healthy children carrying commensal *E. coli* could serve as reservoirs of resistance genes with a possible transmission, if plasmid-mediated, to more virulent bacteria [37,39]. Therefore, future infections in these children could be more challenging to treat if the AMR in commensal *E. coli* is transferred to pathogenic bacteria.

Many of the *E. coli* were MDR, including possible XDR and PDR. This contrast earlier studies from Taiwan [58], China [68], Nigeria [69], and India [53]. The robustness of our study in terms of its considerable sample size and heterogeneous population could have allowed the capture of rare patterns of MDR as compared to the above-mentioned studies. The indicative significant difference in the patterns of MDR and possible XDR between the isolates from diarrhoeic and healthy children is of great public health concern considering the recent report implicating commensal *E. coli* in the maintenance and transfer of XDR plasmid to *Shigella sonnei* which could increase disease morbidity [41].

Importantly, the presence of all possible PDR *E. coli* in healthy children signals a public health threat since the commensal flora is a highly populated ecosystem which may be harbouring bacterial resistant genes and transfer these to other members of the microbiota, including pathogens [70]. This may result in an increased probability of acquiring clinical infections with AMR pathogens. 

Performing risk factor analysis for AMR has been challenging because its occurrence is a complex phenomenon linked to many factors [26,71,72,73]. Some authors have considered analysing only one antimicrobial agent at a time in univariable and rarely on multivariable models [51,74]. The interpretation of such models might not reflect the true situation in which an isolate might have multiple resistance and would not potentially capture the multiple effects of epidemiological factors on the occurrence of AMR. Analysing the association between potential risk factors for AMR must be completed carefully, considering a broad view of the combination of different patterns observed. A multiple resistance patterns approach in a multivariable model is therefore preferable. One of the drawbacks observed in the literature is that many authors discuss results from univariable statistical analyses. In principle, these estimates of Odds Ratio (OR) are unreliable, and more elaborate models need to be developed. 

In the present study, we found a strong communal effect shown through the improved explanatory power of the random-effect models used. The multivariable risk factors analysis revealed an inconsistent association between the multiple resistance patterns outcomes (MDR, possible XDR, possible PDR, and LAMR) and the independent variables. Further, within the subgroups (all children, diarrhoeic and healthy children) of each outcome, no single variable could consistently predict AMR. This variability shows the complexity and heterogeneity of risk factors linked to AMR in the community [11,75,76]. It further implies that the significant variables retained in the final multivariable logistic and ordinal regression models could be a result of random chance and would vary in other communities based on their socio-demographic, economic, behavioural, and environmental characteristics. For instance, four variables in this study, namely residence in a high-density area, feeding the child with stored prepared food, disposing of the solid waste in a pit and by the roadside, predicted the occurrence of MDR in all children in the multivariable logistic model. However, when split between diarrhoeic and healthy children’s subgroups, two different variables (disposing of solid waste in a bin and knowledge of antibiotics) and none of the four from the main model predicted the occurrence of MDR in diarrhoeic children. Equally, one other variable (gender of the child) and only one of the four from the main model predicted the outcome in healthy children. 

Further, after adjusting for the effect of the health centres, which are proxy measures of the residential areas of the study participants, only feeding the child with stored prepared food (adjusted OR: 0.65; *p* = 0.040) remained significantly associated with MDR occurrence in all children. Furthermore, all variables in the diarrhoeic children were insignificant, while gender (adjusted OR: 0.57; *p* = 0.041) and the guardians’ knowledge of antibiotics (adjusted OR: 0.48; *p* = 0.049) remained in the healthy children model. Interestingly, in the ordinal logistic regression model, the level of AMR after adjusting for the effect of the health centres was only significantly influenced by feeding the child with stored prepared food for all children, while in the subgroups, all variables became none-significant. A multivariable analysis and interpretation as shown in this study give a holistic understanding of the interdependence and multiple effects of predictors of the AMR in the community.

Although the large sample size and many antimicrobial agents were used in this study to assess different patterns of AMR, we only used one isolate and one colony from each child’s sample. There is a possibility that some resistant or susceptible colonies were missed, under- or overestimating the true prevalence of resistant *E. coli* and/or certain resistance patterns. Further, linking the potential epidemiological drivers and patterns of AMR to the geographical location of each community should be explored to enable effective surveillance.

## 5. Conclusions

This study has revealed a high occurrence of MDR in healthy and diarrhoeic children with several distinct patterns involving classified antibiotics that are critically important and high-priority antimicrobials for treating serious bacterial infections in humans. It further identified possible PDR—*E. coli* carriage in healthy children, highlighting the role that this group could play in harbouring and transferring resistant genes to other pathogens. 

The limited explanatory power of all logistic models (ROC around 0.60), the different factors popping up in different models, and the strong random effect visualised in the final model point toward a situation where the spread of AMR is linked to the widespread use of antibiotics in the communities involved and cannot be attributed to special exposures. This is also a warning for society, as AMR profiles strongly linked to specific risk factors would be possible to control by focusing on specific hygienic measures within the family unit. If AMR is a community-based problem, interventions also need to be communal—a more fundamental challenge. The considerable sample size of the study should indicate that the patterns observed are representative of the study provinces of Zambia.

## Figures and Tables

**Figure 1 microorganisms-10-01684-f001:**
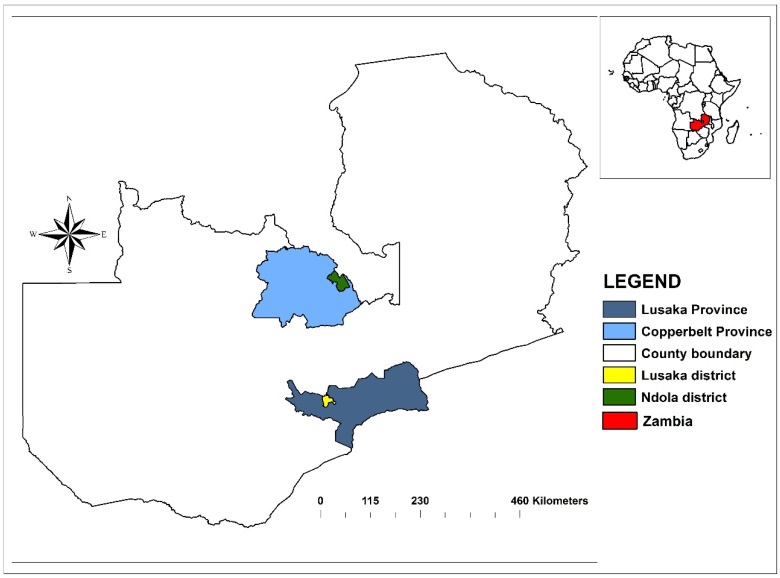
Map of Zambia showing the study sites.

**Figure 2 microorganisms-10-01684-f002:**
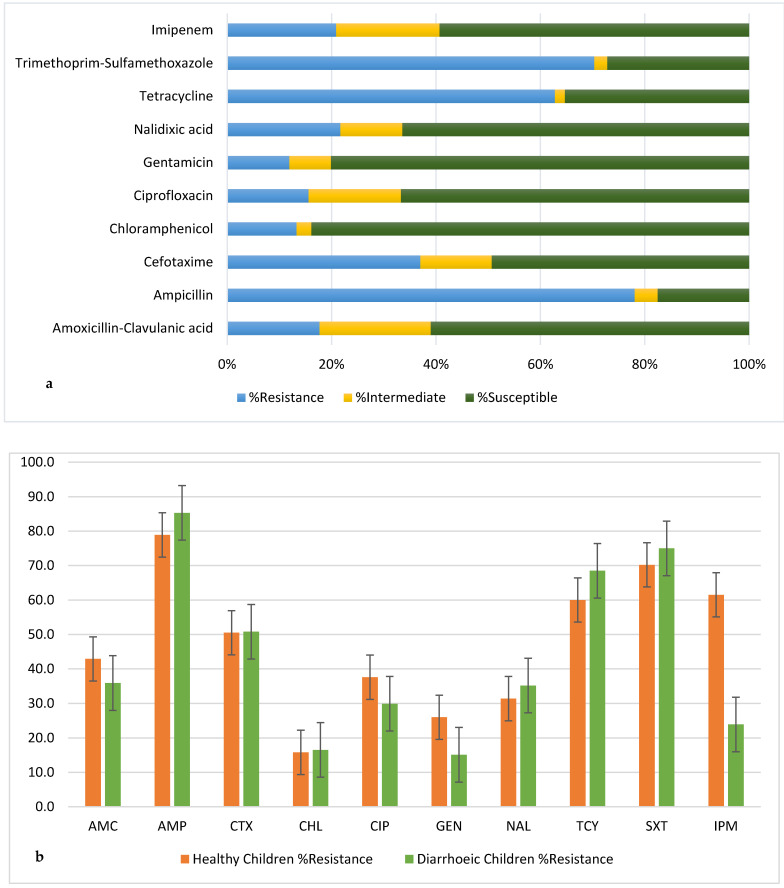
Resistance profile of 1020 *E. coli* isolates; (**a**) Overall and (**b**) healthy versus diarrhoeic children; AMC: Amoxicillin-Clavulanic acid; AMP: Ampicillin; CTX: Cefotaxime; CHL: Chloramphenicol; CIP: Ciprofloxacin; GEN: Gentamicin; NAL: Nalidixic acid; TCY: Tetracycline; SXT: Trimethoprim-Sulfamethoxazole; IPM: Imipenem.

**Figure 3 microorganisms-10-01684-f003:**
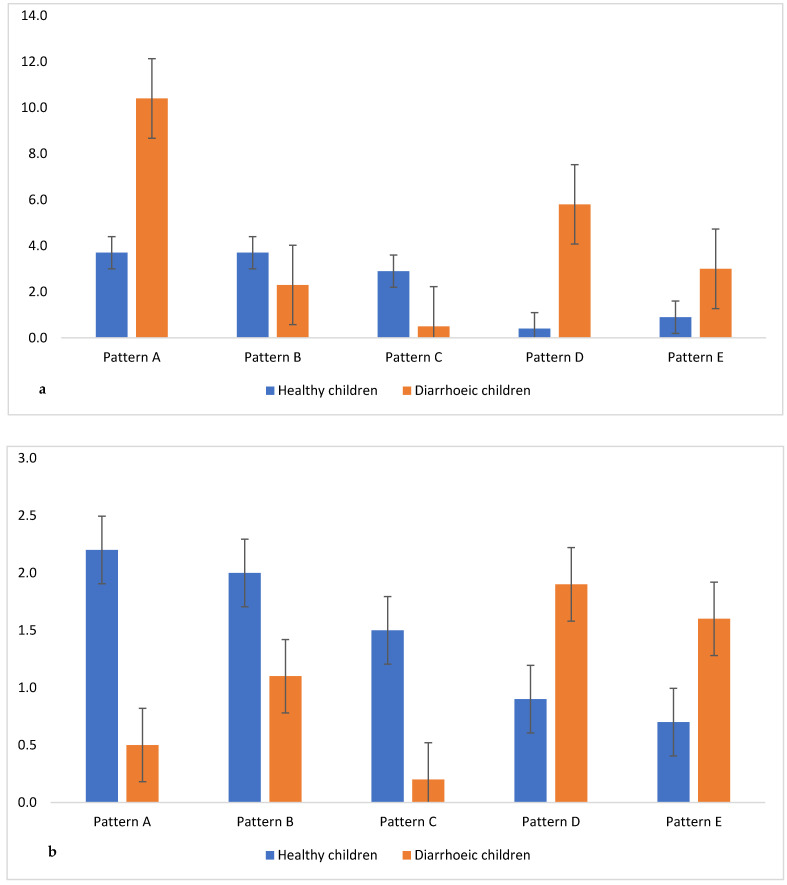
The top five patterns in healthy and diarrhoeic children ((**a**) MDR and (**b**) XDR). **MDR patterns A** (AMP TCY SXT); **B** (AMP TCY SXT IPM); **C** (AMP SXT IPM); **D** (AMP CTX TCY SXT); **E** (AMC AMP CTX TCY SXT); **XDR patterns A** (AMC AMP CTX CIP GEN NAL TCY SXT IPM); **B** (AMC AMP CTX CIP NAL TCY SXT IPM); **C** (AMC AMP CTX CIP TCY SXT IPM); **D** (AMC AMP CTX CHL CIP GEN NAL TCY SXT); **E** (AMC AMP CTX CIP GEN NAL TCY SXT).

**Table 1 microorganisms-10-01684-t001:** List of antibiotics used in AST.

Antibiotics	Class of Antibiotic	Description	Source *
Amoxicillin-clavulanate	Penicillin + beta-lactamase inhibitor	20 µg	Oxoid
Ampicillin	Penicillin (Beta-lactam)	10 µg	Oxoid
Cefotaxime	Third Generation Cephalosporin (Beta-lactam)	30 µg	Oxoid
Chloramphenicol	Phenicols	30 µg	Oxoid
Ciprofloxacin	Fluoroquinolone	5 µg	Oxoid
Gentamicin	Aminoglycosides	10 µg	Oxoid
Nalidixic acid	Quinolones	30 µg	Oxoid
Imipenem	Carbapenems	10 µg	Oxoid
Tetracycline	Tetracycline	30 µg	Oxoid
Trimethoprim-Sulphamethoxazole	Folate Pathway Antagonist	25 µg	Oxoid

* Source: Oxoid, Basingstoke, UK.

**Table 2 microorganisms-10-01684-t002:** Socio-demographic characteristics of study participants.

Variables	Categories	N (1020)	Percent
Health status	Healthy	455	44.61
	Diarrhoeic	565	55.39
Gender	Female	499	48.92
	Male	521	51.08
Age	0–5 months	322	31.57
	6–11 months	239	23.43
	12–35 months	359	58.92
	36–59 months	100	9.80
Guardian’s level of education	None	51	5.00
	Primary	253	24.80
	Secondary	601	30.29
	Tertiary	115	11.27
Population density in the area of habitation	Low density	36	3.53
	Medium density	190	18.63
	High density	794	77.84
Size of the household	Below 5 people	422	41.37
	Equal or above 5 people	598	58.63
Keeping animals at the household level	No	864	84.71
	Yes	156	15.29
Types of animals kept at household level * (N = 156)	Livestock	11	7.05
	Poultry	84	53.85
	Pets	82	52.56
	Other animals	8	5.13

* Variable with multiple responses.

**Table 3 microorganisms-10-01684-t003:** Hygienic and child feeding characteristics of study participants.

Variables	Categories	N (1020)	Percent
Source of water for drinking *	Pipe borne (council water)	859	84.22
	Borehole	147	14.41
	River/Pond/Dam	18	1.76
	Sachet/Bottled/Filtered	11	1.08
Treatment of drinking water	No	366	35.88
	Sometimes	215	21.08
	Yes	439	43.04
Washing hands before cooking and feeding the child	No	51	5.00
	Sometimes	241	23.63
	Yes	728	71.37
Washing hands after disposing of the child’s faeces	No	55	5.39
	Sometimes	182	17.84
	Yes	783	76.76
Types of toilets *	Flush toilet	495	48.53
	Pit latrine	536	52.55
Disposing of solid waste *	Bin	749	73.43
	Pit	238	23.33
	Roadside	36	3.53
Storage of prepared food for the child	No	284	27.84
	Yes	736	72.16
Storage methods of prepared food for the child (N = 736)	At room temperature	331	44.97
	In a cold chain	83	11.28
	In a warmer	322	43.75
Exclusive breastfeeding	No	92	9.02
	Partially	737	72.25
	Exclusively	191	18.73
Child feeding methods *	Spoon	736	72.16
	Fingers/hands	623	61.08
	Bottle feeding	3	0.29

* Variable with multiple responses.

**Table 4 microorganisms-10-01684-t004:** Children’s guardians’ knowledge of antibiotics and characteristics of diarrhoea.

Variables	Categories	N	Percent
Knowledge of antibiotics (N = 1020)	No	845	82.84
	Yes	175	17.16
Correct knowledge of antibiotics by examples (N = 175)	No	65	37.14
	Yes	110	62.86
Awareness of AMR (N = 1020)	No	983	96.37
	Yes	37	3.63
Correct awareness of AMR by the concept definition (N = 37)	No	11	29.73
	Yes	26	70.27
Use of antibiotics suggested by unauthorized personnel	No	740	72.55
	Sometimes	135	13.24
	Yes	145	14.22
Knowledge of causes of diarrhoea (N = 1020)	No	337	33.04
	Yes	683	66.96
Perceived causes of diarrhoea * (N = 683)	Poor hygiene	261	38.21
	Food likely to be contaminated	531	77.75
	Teething	134	19.62
	Undercooked food	18	2.64
	Complimentary food before six months	33	4.83
	Change of diet	5	0.73
	Germs	13	1.90
	Others	26	3.81
Symptoms of diarrhoeic children * (N = 565)	Bloody diarrhoea	33	5.86
	Diarrhoea with mucus	301	53.46
	Fever	347	61.63
	Vomiting	269	47.78

* Variable with multiple responses.

**Table 5 microorganisms-10-01684-t005:** Standard multivariable logistic regression and adjusted for health centres (HC) models for risk factors associated with MDR, possible XDR, and possible PDR in children.

MDR	OR (95% C.I)	*p* > |z|	OR Adjusted for HC (95% C.I)	*p* > |z|
**All children**				
Household in a high-density area	0.13 (0.02–0.99)	0.050		
Storing prepared food for the child	0.65 (0.44–0.96)	0.031	0.65 (0.43–0.98)	0.040
Disposing of solid waste in a pit	1.60 (1.04–2.45)	0.033		
Disposing of solid waste on the roadside	8.80 (1.19–64.96)	0.033		
**Diarrhoeic children**				
Disposing of solid waste in a bin	0.46 (0.27–0.79)	0.006		
Knowledge of antibiotics	1.96 (1.00–3.83)	0.049		
**Healthy children**				
Gender	0.51 (0.31–0.86)	0.011	0.57 (0.32–0.98)	0.041
Storing prepared food for the child	0.60 (0.37–0.98)	0.046		
Knowledge of antibiotics	0.53 (0.28–1.02)	0.057	0.48 (0.23–0.99)	0.049
**Possible XDR**	**OR (95% C.I)**	* **p** * ** > |z|**	**OR adjusted for HC (95% C.I)**	* **p** * ** > |z|**
**All children**				
Age group 6–11 months	0.77 (0.44–1.34)	0.365	0.47 (0.28–0.79)	0.005
Age group 12–35 months	1.02 (0.63–1.65)	0.918	0.55 (0.35–0.88)	0.013
Age group 36–59 months	0.28 (0.11–0.74)	0.010	0.14 (0.05–0.39)	0.000
Child feeding with a spoon	0.61 (0.39–0.95)	0.028		
Size of the household	1.57 (1.08–2.29)	0.019		
**Diarrhoeic children**				
Child feeding with fingers/hands	2.47 (1.02–5.95)	0.044	2.90 (1.16–7.19)	0.022
Keeping poultry at the household level	2.67 (1.18–6.01)	0.018	2.54 (1.09–5.87)	0.029
Using ATB given by non-professional	1.36 (0.98–1.88)	0.058		
**Healthy children**				
Storing prepared food in a warmer	0.48 (0.23–0.99)	0.049		
Keeping pets at household	0.28 (0.08–0.93)	0.038	0.19 (0.05–0.71)	0.013
Treatment of drinking water	1.26 (0.95–1.66)	0.107		
**Possible PDR**	**OR (95% C.I)**	* **p** * ** > |z|**	**OR adjusted for HC (95% C.I)**	* **p** * ** > |z|**
Healthy children				
Awareness of AMR	10.33 (1.93–55.04)	0.006	9.06 (1.48–55.45)	0.017

**Table 6 microorganisms-10-01684-t006:** Ordered logistic regression model for risk factors associated with LAMR in children.

Variables	OR (95% C.I)	*p* > |z|	OR Adjusted for HC (95% C.I)	*p* > |z|
**All children**				
Age group 36–59 months	0.58 (0.34–0.98	0.041		
Household in a medium-density area	0.47 (0.22–0.99)	0.049		
Household in a high-density area	0.48 (0.24–0.98)	0.045		
Storing prepared food for the child	0.61 (0.42–0.87)	0.007	0.65 (0.43–0.98)	0.040
Disposing solid waste in a bin	0.66 (0.48–0.89)	0.006		
**Diarrhoeic children**				
Disposing solid waste in a bin	0.57 (0.38–0.85)	0.006		
Keeping poultry in the household	2.37 (1.17–4.80)	0.016		
Knowledge of antibiotics	1.70 (1.06–2.72)	0.026		
**Healthy children**				
Storing prepared food for the child	0.62 (0.42–0.91)	0.015		
SWH after disposing of the child’s faeces	0.26 (0.08–0.81)	0.021		

SHW: Sometimes Hands washing.

## Data Availability

The essential data supporting the reported results are contained in this study. All other Appendix A can be made available on request from the corresponding author.

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
