# Peer review of "Evidence of Community-Wide Spread of Multi-Drug Resistant Escherichia coli in Young Children in Lusaka and Ndola Districts, Zambia"

_microorganisms, 2022, doi:10.3390/microorganisms10081684_

Round 1

Reviewer 1 Report

1.      Authors should include Bacterial Identification

2.      Authors should explain the Characteristics of E. coli Strains

3.      Authors should include an Analysis of the Antimicrobial Susceptibility

4.      Authors should explain the Clinical Treatment Condition of Zambia With Isolated Strains in the Present Study

5.      Authors should compare the previous studies with current work with a table

Author Response

Thank you for the comments that have been shared with us. We have addressed the reviewer’s comments below:

Response to Reviewer 1 Comments

Point 1: Authors should include Bacterial Identification

Response 1: We acknowledge the comment. Bacterial identification has been added and read as follows “… One colony from each plate was further confirmed by phenotypic characterization and standard biochemical tests using Triple Sugar Iron (Oxoid, Basingstoke, UK), Sulphur Indole Motility (Oxoid, Basingstoke, UK) and Citrate agar (Oxoid, Basingstoke, UK). For additional taxonomic confirmation, 323 isolates were randomly selected for further identification by matrix-assisted laser desorption ionization time-of-flight mass spectrometry (MALDI-TOF MS) using the VITEK® MS – SARAMIS® KB V4.16 (bioMérieux, Lyon, France).”

We have equally added in the results section the following statement: “The identification of isolates using standard biochemical tests was reliable since their confirmation using MALDI-TOF MS showed 98.8% accuracy.”

Point 2: Authors should explain the Characteristics of E. coli Strains

Response 2: We acknowledge the comment. However, the scope of this study was not to characterize the E. coli strains. One of our main interests was to investigate the level (magnitude) of resistance to commonly used antibiotics that could trigger a public health alert since resistance genes can be passed either way among E. coli strains (pathogenic or commensal) hence affecting the management of infectious diarrhoeal diseases.

Point 3: Authors should include an Analysis of the Antimicrobial Susceptibility

Response: We acknowledge the comment. We think that this has been mentioned in the methodology and referenced. We followed the CLSI long-established and known protocol.

Point 4: Authors should explain the Clinical Treatment Condition of Zambia with Isolated Strains in the Present Study

Response: We acknowledge the comment. We have added the following statement: “… However, increased resistance to antibiotics recommended in the Zambia standard treatment guidelines for infectious diseases in children, including diarrhoeal diseases (MoH, 2020), limits treatment options.”

Point 5: Authors should compare the previous studies with current work with a table

Response: We acknowledge the comment. However, this is not an exhaustive review of the existing literature on the field, but a discussion of our results. We have hence done it in text form in the discussion section. We feel that it is normal practice and standard for many journals including MDPI microorganisms.

Reviewer 2 Report

Bumbangi et al., isolated E. coli strains and tested them for antimicrobial resistance (AMR) from 565 diarrhoeic and 455 healthy children.  

  1. I do not agree with the authors in lines 374-381. There was no accurate taxonomic identification used in this study such as 16S rRNA sequencing or targeted PCR for species/AMR. The author used only the traditional culture method for E.coli isolation and AMR typing. I believe there are underrepresentation and/or misrepresentation, and the large sample size does not overweight this potential error. In my opinion, the large sample size rather inflates this type of potential error.
  2. What is the relationship among the 5 patterns? If each pattern is independent, Figure 3 shouldn't be presented as lines (blue and orange). What is the rationale for using lines in figure 3?
  3. I see that the main finding of this manuscript is the IPM differences in Figure 2b. Why don't authors present this separately and highlight it better?
  4. I suggest authors elaborate on AMR found in most healthy children without diarrhoeic symptoms. Being a reservoir can be one as the authors described. How does AMR play a role in the clinical presentation?
  5. Overall, I think the manuscript is unnecessarily lengthy. It can be more concise.

Author Response

Response to Reviewer 2 Comments

Point 1: I do not agree with the authors in lines 374-381. There was no accurate taxonomic identification used in this study such as 16S rRNA sequencing or targeted PCR for species/AMR. The author used only the traditional culture method for E. coli isolation and AMR typing. I believe there are underrepresentation and/or misrepresentation, and the large sample size does not overweight this potential error. In my opinion, the large sample size rather inflates this type of potential error.

Response 1: We acknowledge the comment. We have added in the methodology section the identification of randomly selected E. coli to read “… Later, 323 isolates were randomly selected for further identification by matrix-assisted laser desorption ionization time-of-flight mass spectrometry (MALDI-TOF MS) using the VITEK® MS – SARAMIS® KB V4.16 (bioMérieux, Lyon, France).”

We have equally added in the results section the following statement: “The identification of isolates using standard biochemical tests was reliable since their confirmation using MALDI-TOF MS showed 98.8% accuracy.”

Although we did not carry out16S rRNA sequencing or PCR techniques, the methods used for the identification of E. coli are reliable as shown by MALDI-TOF. The expected error is low, within the acceptable range for any scientific research and cannot change the overall trend reported. Further, the phenotypic characterization and standards biochemical confirmation tests are still acknowledged methods for E. coli as evidenced by their use by other scholars e.g.,

  • Singh AK, (2018). Prevalence of antibiotic resistance in commensal Escherichia coli among the children in rural hill communities of Northeast India. PLoS ONE 13(6): e0199179. https://doi.org/10.1371/journal.pone.0199179
  • GebreSilasie, YM, et al. (2018). Resistance pattern and maternal knowledge, attitude and practices of suspected Diarrheagenic Escherichia coliamong children under 5 years of age in Addis Ababa, Ethiopia: cross-sectional study. BMC Antimicrob Resist Infect Control 7,  https://doi.org/10.1186/s13756-018-0402-5

Nevertheless, the statement: “… However, the large sample size should outweigh some of these risk effects…” has been deleted.

Point 2: What is the relationship among the 5 patterns? If each pattern is independent, Figure 3 shouldn't be presented as lines (blue and orange). What is the rationale for using lines in figure 3?

Response 2: We acknowledge the observation. Indeed, each pattern is independent. We have since changed the figure to the most appropriate one. The figure is depicting the trend for each pattern in healthy and diarrhoeic children.

Point 3: I see that the main finding of this manuscript is the IPM differences in Figure 2b. Why don't authors present this separately and highlight it better?

Response: We acknowledge the comment. We have added an emphasis by incorporating the following statement: “… Nearly 62% and 24% of isolates from healthy and diarrhoeic children were resistant to imipenem, respectively.” The resistance to imipenem has further been highlighted in the discussion.

Point 4: I suggest authors elaborate on AMR found in most healthy children without diarrhoeic symptoms. Being a reservoir can be one as the authors described. How does AMR play a role in the clinical presentation?

Response 4: We acknowledge the comment. We have elaborated in lines 331 – 346. Talking about AMR and its relationship with the clinical presentation might be speculative. Disease dynamics leading to clinical symptoms depend on several factors related to the agent, the host and the environment. However, being a potential reservoir exposes these children to a likely severe disease if infected. Further, if the immunity becomes weak, some commensal organisms may become infectious. This can ultimately challenge the management of the disease using the same antibiotics.

We have therefore added the statement “…Therefore, future infections in these children could possibly be more challenging to treat if the AMR in commensal E. coli is transferred to pathogenic bacteria.”

Point 5: Overall, I think the manuscript is unnecessarily lengthy. It can be more concise.

Response 5: Comment noted, and we have tried to shorten the manuscript.

Reviewer 3 Report

Authors in manuscript presented  occurrence occurrence, patterns and possible drivers of AMR E. coli isolated from children under-five years in Zambia.

The paper is well structured and well written, but there are some minor mistakes that need to be reviewed before publication.

 1.     The Materials and Methods section is too wordy and can be shortened. The writing style should not be for a thesis but rather for a research article - short and concise for the reader. :

a) Please add the classes of antibiotics you tested.
b) I suggest add the company and source after each used media.

2.     The Results section is well-written –  but in Figure 3 please changes values ​​on the chart, so that there is no negative value

3.     The Discussion is well-written.

Additionally, the genus and species of microorganisms must always be written in italics. Throughout the manuscript this rule has often not been respected.

I highly recommend the authors to revise and edit minor mistake. The findings are definitely worth reporting, as long as the format is correct for Microorganisms.

Author Response

Point 1: The Materials and Methods section is too wordy and can be shortened. The writing style should not be for a thesis but rather for a research article - short and concise for the reader:

Response: We acknowledge the concerns, and they have been addressed.

  1. a) Please add the classes of antibiotics you tested.

Response: A table summarizing the requested information has been added.

  1. b) I suggest adding the company and source after each used media.

Response 1: The company and source for each media used have been added accordingly.

Point 2: The Results section is well-written – but in Figure 3 please change values ​​on the chart, so that there is no negative value

 Response 2: The values have been corrected accordingly and there is no negative value anymore.

Point: Additionally, the genus and species of microorganisms must always be written in italics. Throughout the manuscript, this rule has often not been respected.

Response: Genus and species of microorganisms have been written in italics throughout the manuscript.

Round 2

Reviewer 1 Report

Accept in present form

Author Response

Thanks for the comments: Accept in the present form.

We have done the English language editing 

Reviewer 2 Report

The authors' response mentioned corrected figure 3 but I do not see the updated figure reflected in the manuscript. I still see linear lines across 5 patterns which I initially reviewed for authors to revise.

Author Response

1. The authors' response mentioned corrected figure 3 but I do not see the updated figure reflected in the manuscript. I still see linear lines across 5 patterns which I initially reviewed for authors to revise

Response: Thanks for the observation in figure 3. It was an omission, we have removed the linear lines across the 5 patterns. The new figure is between lines 235 and 236.

Response to the general comments: We have done the English language editing.